# In-silico trial of intracranial flow diverters replicates and expands insights from conventional clinical trials

Ali Sarrami-Foroushani [1,2], Toni Lassila [1], Michael MacRaild[1], Joshua Asquith[1], Kit C. B. Roes[3], James V. Byrne [4] & Alejandro F. Frangi [1,2,5,6 ✉]

The cost of clinical trials is ever-increasing. In-silico trials rely on virtual populations and interventions simulated using patient-specific models and may offer a solution to lower these costs. We present the flow diverter performance assessment (FD-PASS) in-silico trial, which models the treatment of intracranial aneurysms in 164 virtual patients with 82 distinct anatomies with a flow-diverting stent, using computational fluid dynamics to quantify post-treatment flow reduction. The predicted FD-PASS flow-diversion success rates replicate the values previously reported in three clinical trials. The in-silico approach allows broader investigation of factors associated with insufficient flow reduction than feasible in a conventional trial. Our findings demonstrate that in-silico trials of endovascular medical devices can: (i) replicate findings of conventional clinical trials, and (ii) perform virtual experiments and sub-group analyses that are difficult or impossible in conventional trials to discover new insights on treatment failure, e.g. in the presence of side-branches or hypertension.

[1] Centre for Computational Imaging and Simulation Technologies in Biomedicine (CISTIB), School of Computing, University of Leeds, Leeds, UK. [2] Leeds Institute for Cardiovascular and Metabolic Medicine (LICAMM), School of Medicine, University of Leeds, Leeds, UK. [3] Department of Health Evidence, Section of Biostatistics, Radboud University Medical Centre, Nijmegen, The Netherlands. [4] Department of Neuroradiology, Nuffield Department of Surgical Sciences, Oxford University, Oxford, UK. [5] Department of Cardiovascular Sciences, KU Leuven, Leuven, Belgium. [6] Department of Electrical Engineering (ESAT), KU Leuven, Leuven, Belgium. ✉email: a.frangi@leeds.ac.uk

Recent developments in patient-specific computational simulations have enabled in-silico trials to predict the safety and efficacy of novel drugs, medical devices or other treatments as part of the research and development life-cycle[1,2]. Some of the benefits of in-silico trials include: (i) enabling more evidence to be obtained before bench or animal studies are started; (ii) extending the trial cohort to rare, extreme or difficult-to-recruit patient phenotypes; (iii) directly comparing two alternative treatments in the same virtual population (reducing the observed effect variance); (iv) evaluating devices under practically challenging physiological conditions that could represent extreme but plausible applications (off-label use); and (v) reducing the number of animals and humans required in trials, and the refinement of long-term studies to minimise suffering[3]. Drug and device regulatory authorities, such as the FDA, are working with the biomedical modelling and simulation community to specify the requirements for introducing evidence obtained from in-silico trials into the regulatory process or informing the design of conventional trials[4].

Leveraging on the recent understanding and developments of in-silico trials, here for the first time, we use an exemplar medical device in-silico trial to: (i) determine whether in-silico trials can replicate outcomes of conventional clinical trials using independent simulated populations that match those of actual clinical trials; and (ii) in the event of successful replication, demonstrate whether such virtual trials can facilitate exploratory virtual experiments not easily achievable in conventional clinical trials, thus providing new insights and generating new hypotheses.

We investigated these aims in the flow diverter (FD) performance assessment (FD-PASS) study, an exemplar in-silico trial for the treatment of intracranial aneurysms using FDs. An FD is a braided, self-expanding, stent-like device implanted through a catheter at the site of a wide-necked cerebral aneurysm. Its function is to reduce blood flow into the aneurysm, instigating the natural formation of an intrasaccular thrombus and blood clotting, eventually leading to a complete eradication of the aneurysm. With over 100,000 cases treated with FDs after their first FDA approval in 2011, these devices revolutionised management of the intracranial aneurysms. Several clinical trials and studies have demonstrated that flow diversion is a safe and effective treatment for many types of aneurysms; however, there are some issues regarding flow diversion failure that have yet to be fully elucidated. Although multiple types of FDs are available in Europe, only the Pipeline Embolization Device (Medtronic) and the Surpass Streamline Flow Diverter (Stryker) are currently FDA approved in the United States[5]. The PED is the oldest and the most well-studied FD and, to date, no other FDs have been shown to outperform the PED. We, therefore, chose to base our in-silico trial on FDs, and the PED in particular, to determine whether FD-PASS can replicate and expand the substantial knowledge acquired from almost a decade of clinical trials and studies on this device.

FDs are primarily used to treat uncoilable, large/giant and wide-neck aneurysms of the internal carotid artery (ICA). However, large/giant aneurysms represent a small fraction of all intracranial aneurysms, as approximately 80% of all unruptured aneurysms are small-/medium-sized ($\leq$10 mm)[6]. Recent trials showed the promise of high occlusion rates in small-/medium-sized, but wide-necked, aneurysms treated with FDs[7,8]. Hence, the FD-PASS trial included wide-neck ($\geq$4 mm), unruptured aneurysms arising from the ICA. This criterion was applied on patient-specific simulation-ready 3D surface models of saccular intracranial aneurysms from the @neurIST project[9] to generate the FD-PASS in-silico trial cohort. FD-PASS primary endpoint was post-treatment aneurysm spatiotemporal mean velocity reduction (AMVR) greater than 35%; which was shown in an

independent population to be an accurate surrogate for complete aneurysm occlusion[10]. Predicted FD efficacy in the FD-PASS trial was compared with the observations in three previously published reference clinical trials (PUFS[11], ASPIRe[7] and PREMIER[8]) to demonstrate whether an in-silico trial can replicate conventional clinical trial findings in independent patient cohorts. The workflow and steps of the FD-PASS in-silico trial are illustrated in Fig. 1.

Although not reported in PUFS, ASPIRe and PREMIER trials, other studies found that a branch artery emerging from the aneurysm sac[12–14], fusiform aneurysm morphology[12], bifurcation aneurysm[12], large/giant aneurysm size[12,15] and age older than 70 years[16,17] were among the risk factors associated with incomplete aneurysm occlusion after flow diversion. The virtual FD-PASS cohort was further stratified into different subgroups to explore factors associated with FD device failure, including the presence of a branch artery arising from the aneurysm and aneurysm sac morphology. Moreover, FD-PASS generated useful evidence beyond that gathered in conventional trials by enabling detailed subgroup analyses to study other risk factors, such as hypertension. There is a growing consensus that assessment of post-flow diversion thrombus composition must predict treatment success in flow diverted aneurysms[18–24]. The FD-PASS in-silico trial demonstrated utility of post-flow diversion thrombosis simulations in individual virtual cases to further explore flow diversion failure scenarios. Such secondary analyses or in-silico experiments would not usually be feasible in a conventional clinical trial, and could contribute to the improved design of medical devices and reduction in adverse events in future.

## Results

**In-silico cohort characteristics.** Eighty-two cases were retained after applying the inclusion and exclusion criteria to the reference @neurIST project population, which provided a sufficiently large sample size according to the power calculations, as described in the Methods section. Three-dimensional anatomic surface model of this cohort was used to build the virtual patient cohort of FD-PASS. Population statistics and measurements of the aneurysms are provided in Table 1, along with a comparison of the in-silico cohort vs. other trial cohorts from the literature. The FD-PASS cohort matched the clinical trial cohorts well in terms of the age and sex distribution, aneurysm location, mean aneurysm dome and neck size. Hypertension was present approximately 50.0% of the cases in the reference clinical trials (Table 1). To understand how hypertension has affected the post-operative occlusion of aneurysms, for each of the 82 virtual anatomies, we simulated two physiological phenotypes, i.e., normotensive and hypertensive, and presented the outcomes, separately, for each group.

**The FD-PASS replicated the efficacy of flow diversion.** The reported success rates of total angiographic occlusion in the in-silico and clinical trials are compared in Table 2. The FD-PASS predicted occlusion rates were 82.9% and 67.1% for normotensive and hypertensive patients, respectively, where an AMVR >35% was considered successful occlusion. Considering the occlusion rates reported by the reference trials within 6–12 months of intervention, the predicted occlusion rate by the FD-PASS replicated the previously reported values.

**FD-PASS allowed subgroup analysis of PED failure scenarios.** Presence of a branch artery originating from the aneurysm sac, large/giant aneurysm size and high aspect ratio (AR) were among the most important risk factors associated with the flow diversion failure[12,25]. The FD-PASS in-silico trial replicated the effect of these morphological risk factors on aneurysm haemodynamics

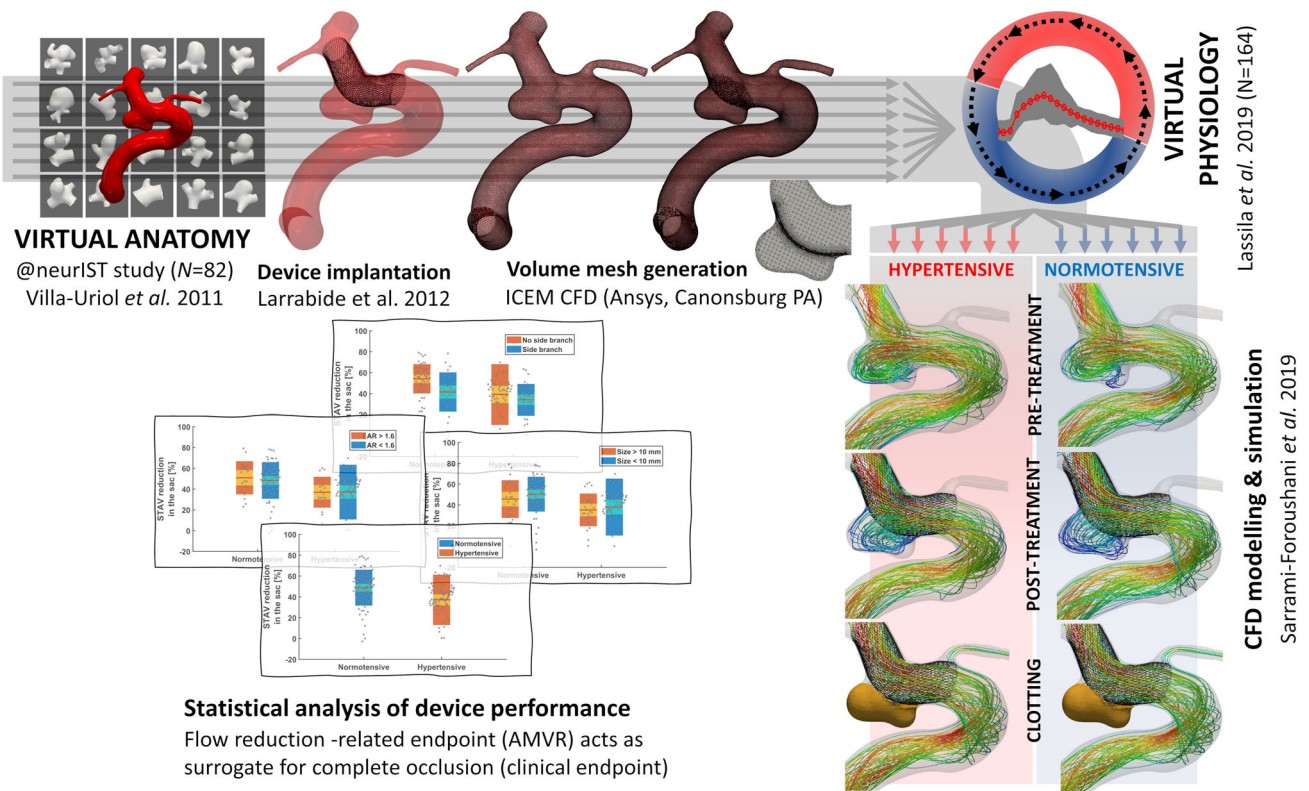

**Fig. 1 Workflow of the FD-PASS in-silico trial.** Virtual anatomies ($N = 82$) are selected from the retrospective cohort of the @neurIST study, following clinical guidelines for flow diversion eligibility. Virtual PED flow diverters are implanted using the algorithm from[38], followed by lumen mesh generation. Carotid flow boundary conditions are generated using the model from[54], representing patients with both normotensive and hypertensive conditions ($N = 164$). The 3-D Navier–Stokes equations are solved and the results post-processed to measure the aneurysm mean velocity reduction (AMVR). A threshold (AMVR >35%) is used to evaluate whether the aneurysm would have occluded completely.

**Table 1 Population characteristics.**

| | FD-PASS in-silico trial | PUFS [29–31] | ASPIRe[7] | PREMIER[8] |
|---|---|---|---|---|
| Number of aneurysms | 82 | 109 | 207 | 141 |
| *Age, year* | | | | |
| Mean ± SD (N) | 54.6 ± 9.6 (82) | 57.0 ± 11.3 (108) | 59.9 ± 12.5 (191) | 54.6 ± 11.3 (141) |
| Median | 50.0 | 59.0 | 60.0 | |
| Range | 36.0–85.0 | 30.2–75.0 | 25.0–89.0 | |
| Female sex, % (n/N) | 78.0 (64/82) | 88.9 (96/108) | 83.8 (160/191) | 87.9 (124/141) |
| Hypertension, % (n/N) | – | 55.6 (60/108) | 53.9 (103/191) | 51.1 (72/141) |
| *Aneurysm size, mm* | | | | |
| Mean ± SD (N) | 9.1 ± 4.0 (82) | 18.2 ± 6.5 (82) | 14.5 ± 6.9 (207) | 5.0 ± 1.9 (141) |
| Median | 8.0 | 17.5 | 12.0 | 4.6 |
| Range | 3.5–25.5 | 6.2–36.1 | 0.9–41.0 | |
| *Aneurysm neck, mm* | | | | |
| Mean ± SD (N) | 5.5 ± 1.4 (82) | 8.8 ± 4.3 (108) | 7.1 ± 4.2 (202) | 4.0 ± 1.4 (141) |
| Median | 5.0 | 8.1 | 6.0 | 3.7 |
| Range | 4.0–9.8 | 4.1–36.1 | 0.8–53.0 | |
| *Aneurysm location* | | | | |
| ICA/PCoA, % (n/N) | 100 (82/82) | 100 (108/108) | 90.8 (188/207) | 95 (134/141) |
| Multiple PEDs used, % (n/N) | 0.0 (0/82) | 98.1 (105/107) | 18.8 (39/207) | 6.4 (9/141) |
| Efficacy endpoint | AMVR >35% as a proxy for angiographic occlusion | Angiographic occlusion | Angiographic occlusion | Angiographic occlusion |

*ICA* internal carotid artery, *PCoA* posterior communicating artery, *PED* pipeline embolisation device.

and occlusion (i.e., device failure). Aneurysm haemodynamics was assessed before and after flow diversion by measuring the sac space-and-time averaged velocity (STAV), neck maximum time-averaged velocity (MTAV) and post-flow diversion reduction in STAV (i.e., AMVR) and MTAV. The virtual FD-PASS cohort was further stratified, and these haemodynamic variables were compared between the subgroups of interest to explore the association of the FD device failure with each of the risk factors, i.e., presence

of a branch artery, aneurysm size and AR (Fig. 2: first to third row and Supplementary Tables 1–3).

**PED failure in the presence of a branch artery.** Among 82 virtual anatomies in the FD-PASS, 34 (41.5%) had branch arteries originating from the aneurysm sac. Patient demographic and aneurysm morphology characteristics did not significantly differ between aneurysms with and those without a side branch

**Table 2 Flow diversion efficacy: angiographic flow reduction in the in-silico trial compared to measures of flow diversion efficacy in three reference clinical trials.**

|  | FD-PASS in-silico trial | PUFS[29–31] | ASPIRe[7] | PREMIER[8] |
|---|---|---|---|---|
| Number of aneurysms | 82 (NT) 82 (HT) | 109 | 207 | 141 |
| FD device | PED | PED | PED | PED |
| Flow diversion efficacy | 82.9% (NT) 67.1% (HT) | 73.6% (at 6 months) 86.8% (at 1 year) 95.2% (at 5 years) | 74.8% (at 7.8 months) | 76.8% (at 1 year) |

*FD* flow diverter, *NT* normotensive patients, *HT* hypertensive patients, *PED* pipeline embolisation device.

**Table 3 Demographics and morphological characteristics of aneurysms with/without a side branch.**

|  | With side branch | Without side branch | *p* value |
|---|---|---|---|
| *Number of aneurysms* | 34 | 48 |  |
| *Age, year* |  |  |  |
| Mean ± SD | 55.0 ± 9.4 | 54.4 ± 9.9 | 0.78 |
| Median | 51.0 | 50.0 |  |
| Range | 36.0–77.0 | 36.0–85.0 |  |
| *Female sex, % (n/N)* | 82.3 (28/34) | 75.0 (36/48) | 0.43 |
| *Aneurysm size, mm* |  |  |  |
| Mean ± SD | 8.9 ± 3.1 | 9.1 ± 4.5 | 0.79 |
| Median | 8.3 | 7.8 |  |
| Range | 5.0–18.9 | 3.5–25.5 |  |
| *Aneurysm neck, mm* |  |  |  |
| Mean ± SD | 5.6 ± 1.4 | 5.4 ± 1.3 | 0.54 |
| Median | 5.2 | 4.5 |  |
| Range | 4.1–9.3 | 4.0–9.8 |  |
| *Non-sphericity* |  |  |  |
| Mean ± SD | 0.17 ± 0.06 | 0.15 ± 0.08 | 0.27 |
| Median | 0.17 | 0.16 |  |
| Range | 0.05–0.30 | 0.03–0.35 |  |

*p* values were computed using two-tailed *t*-tests.

(Table 3). No significant differences were observed in aneurysm haemodynamics before flow diversion (Supplementary Table 1). However, the post-treatment reductions in both STAV and MTAV were significantly higher in aneurysms without a side branch ($p = 0.009$ and 0.002, respectively; Fig. 2: first row; Supplementary Table 1). Occlusion rates, determined by AMVR, were 70.6% and 91.7% for normotensive aneurysms with and without side branches, respectively ($p < 0.05$; Table 4). These reduced to 52.9% and 77.1% for hypertensive aneurysms with and without side branches, respectively ($p < 0.05$; Table 4). The virtual flow simulation indicates that a side branch is likely to cause higher residual flow within the aneurysm sac.

**PED failure in large/giant and high aspect ratio aneurysms.** Among 82 virtual patients in the FD-PASS cohort, 26 (31.7%) patients had large/giant aneurysms (diameter >10 mm). The demographic characteristics did not significantly differ between the large/giant and small aneurysm subgroups (Table 5). While the pre-treatment aneurysm STAV was significantly higher in the small aneurysm subgroup ($p < 0.001$), the post-treatment reduction in the sac STAV (AMVR) did not significantly differ between the two subgroups (Fig. 2: second row; Supplementary Table 2). Occlusion rates, determined by AMVR, were 73.1% and 87.5% for normotensive large/giant and small aneurysms, respectively (Table 4, $p \approx 0.1$). These reduced to 50.0% and 75.0% for hypertensive large/giant and small aneurysms, respectively, ($p < 0.05$; Table 4).

Among 82 virtual patients in the FD-PASS cohort, 18 (21.9%) patients had aneurysms with an AR greater than 1.6. The demographic characteristics did not significantly differ between the subgroups with high and low AR aneurysms (Table 5). However, the aneurysms were significantly larger in the high AR

subgroup ($p < 0.001$), and the pre-treatment aneurysm STAV was significantly higher in the low AR aneurysm (AR < 1.6) subgroup ($p < 0.001$). Even so, the post-treatment reduction in the sac STAV (AMVR) or MTAV did not significantly differ between the two subgroups (Fig. 2: third row; Supplementary Table 3). Occlusion rates with respect to the AMVR were 77.8% and 84.4% for the normotensive high and low AR aneurysms, respectively (Table 4). These reduced to 55.6% and 70.3% for hypertensive high and low AR aneurysms, respectively (Table 4). These differences were not statistically significant ($p > 0.05$). Taken together, from a purely haemodynamic viewpoint, large/giant aneurysms or aneurysms with high ARs did not respond differently to flow diversion compared to small aneurysms or aneurysms with lower ARs, respectively.

**FD-PASS expanded exploratory analyses of clinical trials.** One benefit of in-silico trials is the practicality to perform exploratory experiments to provide new insights and generate new hypotheses. PED flow diversion efficacy in the presence of hypertension has been clinically less studied so far. The in-silico nature of FD-PASS allowed us to perform normotensive and hypertensive simulations for each of the 82 virtual anatomies. We computationally estimated reductions in STAV (i.e., AMVR) and MTAV after flow diversion and compared them between the normotensive and hypertensive subgroups (Fig. 2: fourth row and Supplementary Table 4). No significant differences were observed in aneurysm haemodynamics before flow diversion (Supplementary Table 4). However, the post-treatment reductions in both STAV and MTAV were significantly higher in normotensive patients ($p < 0.001$ and $p = 0.007$, respectively; Fig. 2: fourth row; Supplementary Table 4). Occlusion rates, determined by AMVR, were 82.9% and 67.1% for normotensive and hypertensive patients, respectively ($p < 0.05$; Table 4). Our simulation results clearly show that hypertension is likely to cause less effective flow diversion. We argue that this observation demonstrates how the FD-PASS can expand insights from conventional clinical trials and be used to generate hypotheses or priors that can augment later clinical trials.

Although none of the three reference clinical trials reported a significant association between aneurysm morphology and occlusion rates, several other clinical studies have reported cases of haemorrhagic stroke due to the delayed rupture of complex-shaped aneurysms after flow diversion[15,20]. In addition, the IntrePED study reported an association between hypertension and ischaemic stroke in flow-diverted aneurysms[26]. While the ability to investigate such scenarios is limited in real patient cohorts, the in-silico nature of FD-PASS allows us to apply advanced modelling techniques to allow the generation and examination of clinical hypotheses about these observed complications. To achieve this aim, we simulated post-flow diversion clot formation in our virtual patients with ICA aneurysms. The clot formation model and its validation are detailed in[27], and all the model parameters used were the same for all virtual patients. The clot formation model was simulated until a stable steady-state was reached, after which the stability of the virtual

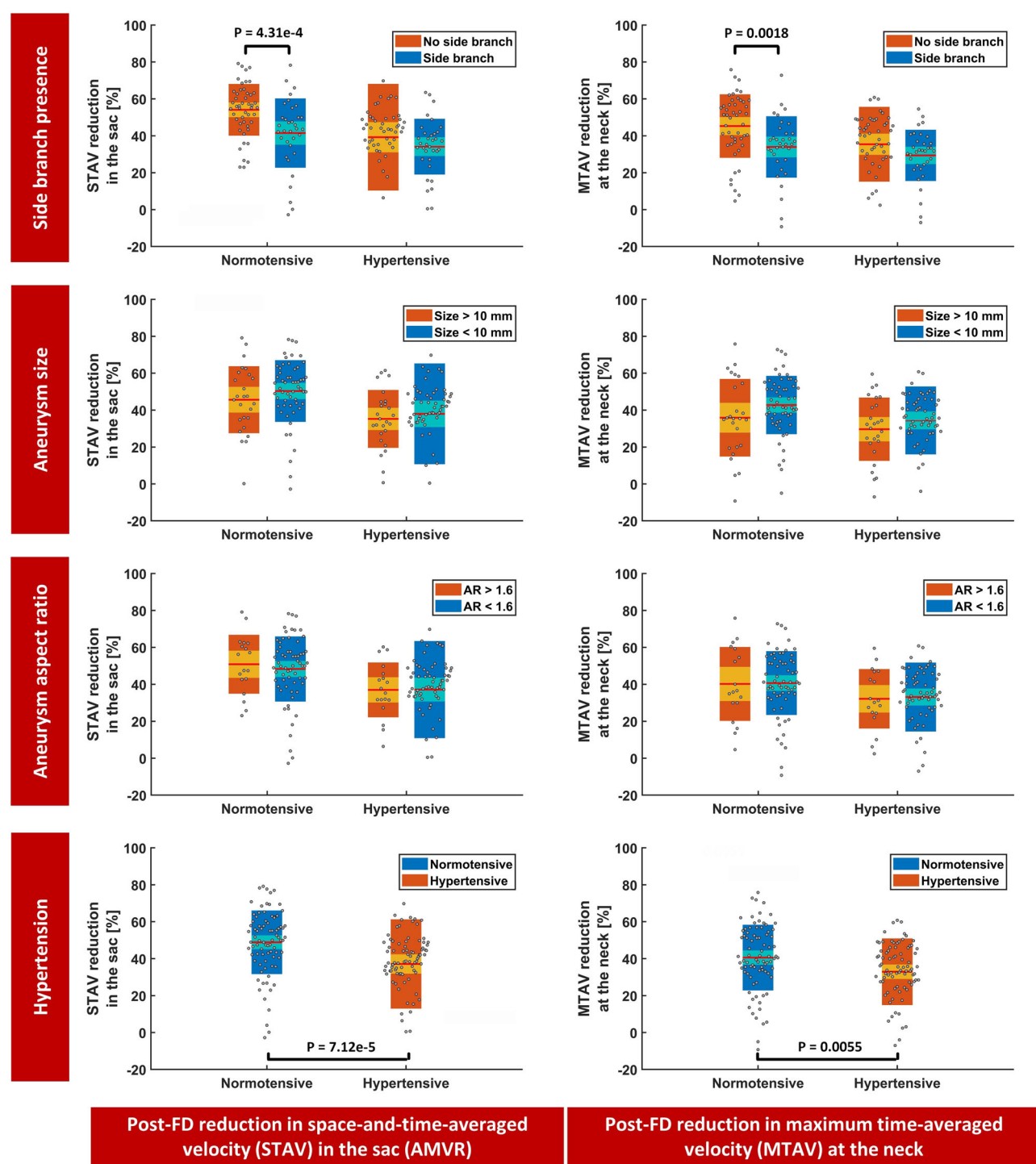

**Fig. 2 Haemodynamic performance of flow diversion in FD-PASS.** Comparison of haemodynamic performance of the PED in aneurysms with and without a branch artery emerging from the sac (first row), aneurysms with a diameter <10 mm (small) and >10 mm (large/giant aneurysms) (second row), aneurysms with a high and low AR (third row) and hypertensive and normotensive aneurysms (fourth row). Left column: post-flow diversion reduction in the space-and-time-averaged velocity (STAV) in the aneurysm sac. Right column: post-flow diversion reduction in the maximum time-averaged velocity (MTAV) entering the aneurysm at the neck. The results are shown for normotensive patients ($N = 82$), and hypertensive patients ($N = 82$). STAV space-and-time-averaged velocity, MTAV maximum time-averaged velocity, AR aspect ratio. The central mark denotes the mean, the smaller box denotes 1.96 standard error of the mean (SEM) and the larger box denotes the standard deviation (SD). p values were computed from one-tailed t-tests (unpaired).

clot was evaluated using the flow-induced platelet index (FiPi). Compared with experimental results, FiPi values <0.15 were considered indicative of unstable red thrombi[27], which have been linked to post-operative aneurysm rupture even in the presence of complete occlusion[20]. Predictions were insensitive to changes in this threshold of up to 30%. Therefore, we used FiPi <0.15 as a data-independent threshold to identify red thrombi in the FD-PASS data.

**Table 4 Subgroup analysis: the effects of the presence of a side branch, aneurysm size, aspect ratio and hypertension on flow diversion efficacy (determined by AMVR).**

| | Normotensive | | | Hypertensive | | |
|---|---|---|---|---|---|---|
| | SO/subtotal | Difference (95% CI) | *p* value | SO/subtotal | Difference (95% CI) | *p* value |
| *Side branch* | | | | | | |
| Yes | 70.6% (24/34) | 21.1% (4.2 to 38.6%) | 0.013 | 52.9% (18/34) | 24.2% (3.4 to 43.0%) | 0.023 |
| No | 91.7% (44/48) | Ref | Ref | 77.1% (37/48) | Ref | Ref |
| *Size* | | | | | | |
| >10 mm | 73.1% (19/26) | 14.4% (−2.8 to 34.6%) | 0.108 | 50.0% (13/26) | 25.0% (3.0 to 45.3%) | 0.026 |
| <10 mm | 87.5% (49/56) | Ref | Ref | 75.0% (42/56) | Ref | Ref |
| *Aspect ratio* | | | | | | |
| >1.6 | 77.8% (14/18) | 6.7% (−10.5 to 30.6%) | 0.514 | 55.6% (10/18) | 14.7% (−8.5 to 38.7%) | 0.242 |
| <1.6 | 84.4% (54/64) | Ref | Ref | 70.3% (45/64) | Ref | Ref |
| *Hypertension* | 82.9% (68/82) | Ref | Ref | 67.1% (55/82) | 15.8% (2.6 to 28.4%) | 0.0196 |

*p* values were computed using the χ² test.
*AMVR* aneurysm mean (time-averaged) velocity reduction, *SO* successful occlusion indicated by AMVR, *Ref* group taken as reference.

---

**Table 5 Demographics and morphological characteristics of small/large and high/low aspect ratio aneurysms.**

| | Size >10 mm | Size <10 mm | *p* value | AR >1.6 | AR <1.6 | *p* value |
|---|---|---|---|---|---|---|
| *Number of aneurysms* | 52 | 112 | | 36 | 128 | |
| *Age, year* | | | | | | |
| Mean ± SD | 52.1 ± 9.2 | 55.8 ± 9.7 | 0.10 | 51.2 ± 7.9 | 55.6 ± 9.9 | 0.10 |
| Median | 50.0 | 52.0 | | 50.0 | 50.0 | |
| Range | 36.0–77.0 | 36.0–85.0 | | 40.0–74.0 | 36.0–85.0 | |
| *Female sex, % (n/N)* | 73.1 (38/52) | 80.36 (90/112) | 0.46 | 83.4 (30/36) | 76.6 (98/128) | 0.55 |
| *Aneurysm size, mm* | | | | | | |
| Mean ± SD | 13.6 ± 3.9 | 6.9 ± 1.6 | 4.0e-34 | 14.2 ± 4.5 | 7.6 ± 2.3 | 2.7e-24 |
| Median | 11.8 | 7.0 | | 12.9 | 7.2 | |
| Range | 10.1–25.5 | 3.5–9.9 | | 9.7–25.5 | 3.5–15.2 | |
| *Aneurysm neck, mm* | | | | | | |
| Mean ± SD | 6.7 ± 1.6 | 4.9 ± 0.8 | 3.0e-17 | 5.8 ± 1.4 | 5.4 ± 1.4 | 0.064 |
| Median | 6.6 | 4.7 | | 5.6 | 5.0 | |
| Range | 4.3–9.8 | 4.0–7.5 | | 4.1–8.4 | 4.0–9.8 | |
| *Non-sphericity* | | | | | | |
| Mean ± SD | 0.20 ± 0.06 | 0.14 ± 0.07 | 4.1e-6 | 0.24 ± .04 | 0.14 ± 0.06 | 3.8e-15 |
| Median | 0.21 | 0.15 | | 0.23 | 0.14 | |
| Range | 0.06–0.35 | 0.03–0.30 | | 0.16–0.35 | 0.03–0.29 | |

*p* values were computed using the two-tailed *t*-test.
*SB* side branch, *STAV* space-and-time-averaged velocity, *MTAV* maximum time-averaged velocity.

**Haemorrhagic stroke in large complex-shaped aneurysms**. We simulated post-treatment clotting in two virtual patients: one with a typical simple aneurysm (Fig. 3, case 1, size = 6.1 mm, AR = 0.72, non-sphericity = 0.08) and one with a complex-shaped large aneurysm (Fig. 3, case 2, size = 16.7 mm, AR = 1.9, non-sphericity = 0.25). Both aneurysms clotted within a similar time scale. The clot that formed in the simple aneurysm (case 1) was an organised white thrombus with high platelet content throughout the formation and grew in a layer-wise manner. In the more complex case 2, we observed the rapid formation of a non-organised red thrombus (FiPi < 0.15) inside the aneurysm bleb. A persistent flow jet was present in the sac during the entire time course of the clot formation. It has been hypothesised that the co-presence of unorganised red thrombi and a persisting flow jet in complex-shaped flow-diverted aneurysms can cause the delayed rupture of aneurysms and haemorrhagic stroke[20]. The IntrePED study also reported an association between large aneurysm size and haemorrhagic stroke after flow diversion[15]. Our observations demonstrate how the FD-PASS can test experimental scenarios to explore why FDs fail in complex-shaped aneurysms.

**Ischaemic stroke in hypertensive patients**. Next, we simulated post-treatment clotting in two virtual patients with precisely the same aneurysm morphology (Fig. 3, cases 3 and 4, size = 7.2 mm,

AR = 1.25, non-sphericity = 0.21), but differing blood pressure levels (one normotensive and the other hypertensive). We observed the rapid formation of non-organised thrombi in the aneurysm bleb. However, in the normotensive case (case 3), the course of clot formation continued with the growth of an organised white thrombus that filled the aneurysm sac, keeping the ophthalmic artery patent. The entire course of thrombosis was almost ten-times faster than that in cases 1 and 2, so this aneurysm may occlude fast enough and the presence of a red thrombus in the bleb would not result in aneurysm rupture, as the organised thrombus filled the aneurysm and prevented blood flow into the sac. In hypertensive case 4, which was hypertensive, however, we observed patency of the aneurysm core and the transport/formation of a red thrombus in the branching posterior communicating artery (PCoA), which could cause a stroke in the downstream vessel. Ischaemic stroke has been reported in hypertensive patients treated with FDs[26]. We argue that these observations provide another demonstration of how the FD-PASS can be used for virtual experimentation to explain concepts that are difficult to study in conventional clinical trials, such as why ischaemia occurs in such patients.

**Discussion**

Viceconti et al.[28] defined three principles that every in-silico trial should follow to be credibly comparable to a conventional clinical

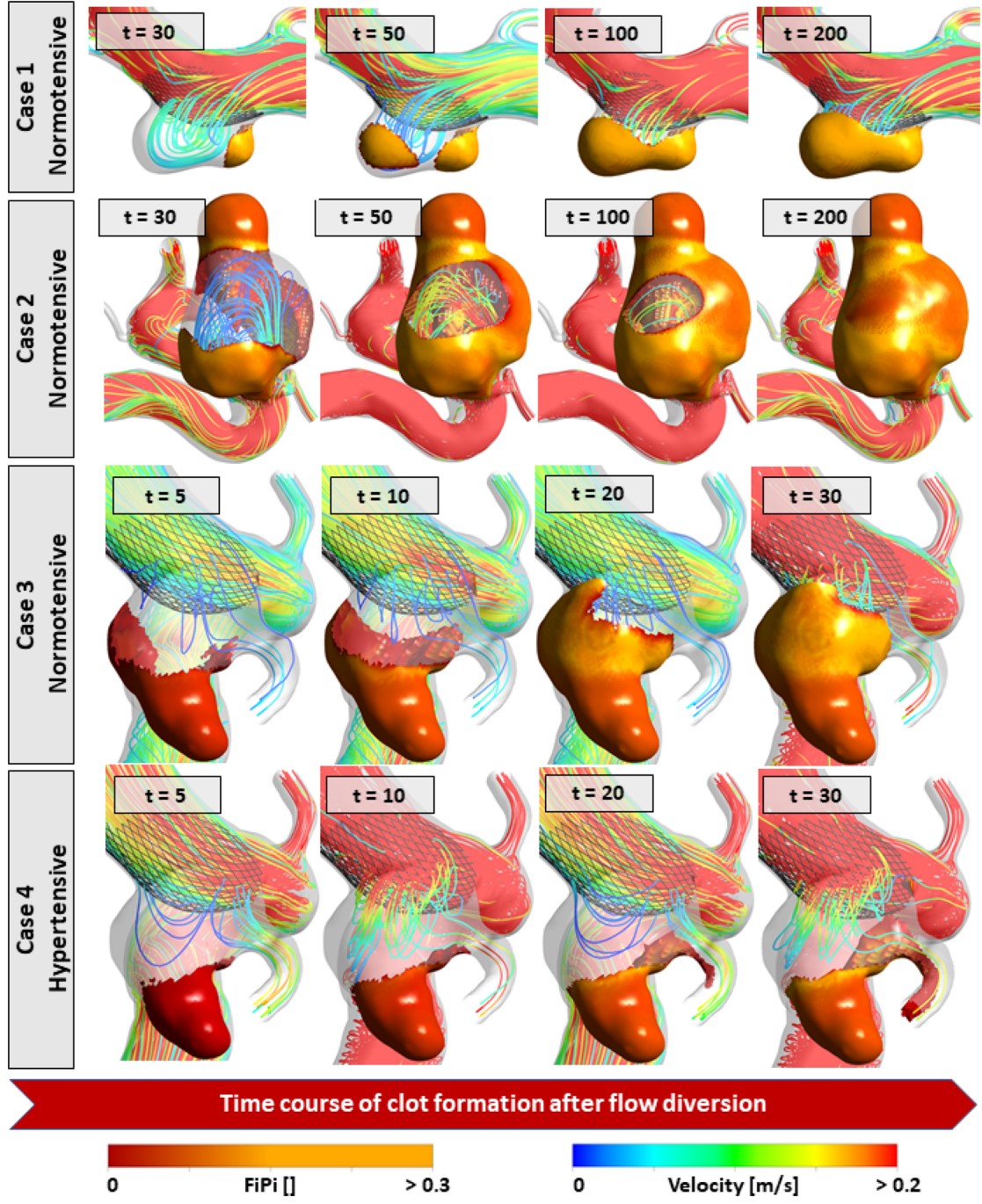

**Fig. 3 Time course of post-treatment clot formation in four virtual patients.** The flow-induced platelet index (FiPi) is a measure of the clot platelet content and indicates whether the clot is a red thrombus (FiPi < 0.15) or white thrombus (FiPi > 0.15). White thrombi have higher stability than red thrombi. Red thrombi are also known to induce autolytic activities in the wall, resulting in the weakening of the wall and aneurysm rupture. Time values are shown in units of simulation time and are not representative of physiological time scales but are reported to enable comparisons between patients.

trial: (i) every virtual patient has to be plausible when compared to patients in a clinical trial cohort, (ii) the accuracy of the virtual endpoint should represent the accuracy of the endpoint in the clinical trial cohort and (iii) for sufficiently large physical and virtual cohorts, the conclusions made from observed effects should be identical. The FD-PASS in-silico trial satisfied these three principles in the following ways.

(i) The virtual patients were simulated from real, patient-specific imaging data, and virtual patient selection was performed following the same inclusion/exclusion criteria as the reference clinical trials. This allowed the direct comparison between the

demographic and anatomical characteristics of virtual and real patient cohorts. All virtual patient anatomies in the FD-PASS were built based on the @neurIST 3D rotational angiography imaging cohort with appropriate inclusion/exclusion criteria applied to select the virtual cohort. Therefore, the virtual patients simulated here were entirely independent of the actual patients in the reference clinical trials.

(ii) Validated computational models were used, and we ensured that the endpoints of the in-silico trial followed a similar distribution as the clinical endpoints. The computational in-silico trial setup of the FD-PASS trial was an orchestration of building

blocks that were presumed to be adequately validated in previous studies (see Methods). What distinguishes the concept of validation of this in-silico trial from that in personalised medicine or patient-specific modelling paradigms is that, here, the underlying models and simulations were not tailored for a specific patient but should be able to represent medical device performance over an entire population of virtual patients.

(iii) We demonstrated that similar quantitative conclusions can be made about the performance of FDs in both in-silico trials and clinical trials. This was achieved in the FD-PASS by showing that the predicted post-operative occlusion likelihood were similar to results observed in the reference clinical trials.

Three major clinical trials on PED FDs, i.e., PUFS, ASPIRe and PREMIER, reported angiographic occlusion rates of 74–77% at approximately 1 year after flow diversion[7,8,29–31]. Large meta-analyses of flow diversion efficacy confirmed similar occlusion rates within 6–12 months after treatment[7]. Longer-term occlusion, however, increased up to 95%, as reported at 5 years in the PUFS[31]. Given the ideal conditions of trials such as PUFS, in[16], it was argued that the overall long-term occlusion rate after PED treatment could reach 85%. The FD-PASS in-silico trial estimated an occlusion rate of 82.9% in patients with normal blood pressure and 67% in hypertensive patients, which replicated the occlusion rates reported in the reference clinical trials. FD-PASS estimation of occlusion rate was performed based on one primary endpoint, the post-treatment AMVR >35% as a surrogate for angiographic occlusion, which was obtained by computational simulation of an independent cohort of $N = 12$ flow diverted aneurysms[10]. While we emphasise that FD-PASS did not aim to devise or validate a predictive surrogate endpoint, we explored what happens to the occlusion rates when we set a different surrogate endpoint. We examined AMVR >20% and post-treatment inflow reduction (INFR) at the aneurysm neck >30%, which were presented as accurate simulation-based predictors of flow diversion success in a cohort of 36 patients treated with intrasaccular FDs[32]. Occlusion rates of 78% and 83% were achieved when we used AMVR >20% and INFR >30% as primary endpoints, respectively. This demonstrated that the estimated occlusion rates were not sensitive to the choice of surrogate endpoint.

Although not reported in PUFS, ASPIRe and PREMIER trials, other studies listed a branch artery emerging from the aneurysm sac[12–14], large/giant aneurysm size[12,15] among the risk factors associated with incomplete aneurysm occlusion after flow diversion. In-silico trials can offer detailed subgroup analysis of the outcomes. On univariate analysis of results presented in[4], FD-PASS showed higher risks of incomplete occlusion in aneurysms with a branch vessel (risk ratio (RR): 3.53; CI: 1.21–10.32; $p = 0.021$) but not in aneurysms with size >10 mm (RR: 2.15; CI: 0.84–5.51; $p = 0.109$) (Table 6).

The association of hypertension with incomplete aneurysm occlusion was also not reported in the comparative human trials, presumably due to the impracticality of recruiting a balanced cohort of patients to allow an efficient assessment of device performance under different physiological conditions. The FD-PASS trial allowed the comparison of hyper- vs. normotensive patients, which is challenging to investigate with human participants, thus generating useful evidence beyond that gathered in conventional trials. For each virtual patient, in whom we were able to maintain the anatomy and the deployed device configuration, we studied the post-treatment haemodynamics with two different physiological flow conditions (normotensive and hypertensive). On Univariate analysis, we observed higher risks of incomplete occlusion in hypertensive patients (RR: 1.93; CI: 1.09–3.40; $p = 0.023$) (Table 6). Such control of sources of variability are not readily available in conventional clinical trials.

The IntrePED registry[15,26,33] found higher risks of ischaemic and haemorrhagic stroke in hypertensive patients and in patients with large/giant aneurysms, respectively. To date, clinical trials have been limited in their ability to identify the underlying mechanisms relating to the increased stroke risk. For instance, based on observations in three flow-diverted aneurysms, Kulcsár et al.[20] hypothesised that the co-presence of non-organised red thrombi and a persistent flow jet could be associated with post-treatment rupture in large and complex-shaped aneurysms. The likelihood of such complications cannot be assessed merely based on haemodynamic simulations. In our small-scale secondary analysis of the FD-PASS cohort, thrombosis formation was simulated in four patients treated with a PED: one with a small aneurysm, one with a large complex-shaped aneurysm, and two with the same anatomy but different flow conditions (hypertensive and normotensive). We observed deposition of the clot into the PCoA in the hypertensive patient, which may explain the ischaemic stroke event in this patient. We also observed the co-presence of an entering jet and non-organised red thrombi in the complex-shaped aneurysm, which may explain the elevated risk of post-treatment rupture as reported in[20]. This provides a demonstration of how the in-silico trials can utilise advanced modelling and simulation techniques for virtual experimentation of scenarios that are impossible to study in conventional clinical trials, such as investigation of clot composition and stability after flow diversion.

The findings from the FD-PASS in-silico trial replicated those of previously published conventional clinical trials. This study enabled further stratification of the virtual patient cohort to assess efficacy in different subgroups. We argue that although large-scale studies are required for the general acceptance of medical devices and the overall evaluation of associated risks, as recommended in[17,34], treatment planning for devices with complex mechanisms of action (such as the PED) needs to be determined individually for each patient. The FD-PASS trial offered additional information about populations more likely to experience device failure that would not usually be available from a conventional clinical trial. We demonstrated the use of advanced modelling and simulation techniques to explain the underlying mechanisms of complications and to advise clinical decisions on a case-by-case basis. Moreover, while issues related to device sizing,

**Table 6 Subgroup analysis: RRs of incomplete occlusion associated with the presence of a side branch, aneurysm size, aspect ratio and hypertension.**

| | Normotensive | | Hypertensive | | | |
|---|---|---|---|---|---|---|
| | Relative risk (95% CI) | *p* value | Relative risk (95% CI) | *p* value | Relative risk (95% CI) | *p* value |
| Side branch presence | 3.53 (1.21–10.32) | 0.021 | 2.05 (1.09–3.85) | 0.025 | | |
| Size (>10 mm) | 2.15 (0.84–5.51) | 0.109 | 2.00 (1.10–3.62) | 0.022 | | |
| Aspect ratio (>1.6) | 1.42 (0.50–4.00) | 0.504 | 1.49 (0.79–2.83) | 0.216 | | |
| Hypertension | | | | | 1.93 (1.09–3.40) | 0.023 |

*p* values were computed using the two-tailed *t*-test.

positioning, deployment and apposition with the vessel wall were not considered in the FD-PASS trial, the in-silico clinical trial approach does also offer benefits in this respect because multiple device deployments can be performed on the same patient to compare different deployment strategies. Were a more detailed deployment model implemented, the FD-PASS trial could also be extended to perform such comparisons. Future work will involve setting up a prospective digital twin study, where patients treated with the PED are followed, and aneurysm occlusion and thrombus formation are monitored by angiographic imaging after 12 months. Such a study would allow more comprehensive in-vivo validation of the predictions of our thrombus formation model.

## Methods

**FD-PASS in-silico trial design**. A population of 301 patient-specific simulation-ready surface models of saccular intracranial aneurysms from the @neurIST project[9] was the reference population in this study. The mean age of the population was $51.2 \pm 8.2$ years, and 81.2% were female. The mean aneurysm size was $7.4 \pm 3.9$ mm (range 2.0–31.0). The mean aneurysm neck size was $4.6 \pm 1.7$ mm (range 1.7–12.2). Aneurysms, if treated, were coiled, and treatment outcomes were not available for this population. However, as the primary aim of the FD-PASS trial was to perform an in-silico replication of outcomes from three previously published flow diversion clinical trials in an independent population, a direct comparison of the treatment outcomes between the real @neurIST project cases and their digital twins was not of interest. Therefore, we used the 301 untreated aneurysms as our reference population.

**Inclusion criteria and virtual patient selection**. The inclusion criteria in this study were as follows: wide-neck ($\geq 4$ mm), unruptured aneurysm arising from the ICA (cavernous through the anterior choroidal segments). These criteria based on conventional clinical trials such as PUFS[30] and PREMIER[8]. Aneurysm dimensions were measured on the virtual anatomies obtained from 3D rotational angiography images. The primary cause of exclusion was poor image resolution that prevented the creation of virtual anatomy models amenable to computational simulations. Eighty-two cases were retained after applying the inclusion and exclusion criteria to the reference cohort, which provided a sufficiently large sample size according to the power calculations, as described in the next section.

For each of the 82 virtual anatomies, we simulated two physiological phenotypes, i.e., normotensive and hypertensive. To do this, we used a statistical model for cerebral flow variability (see[35] for model details) to generate both normotensive and hypertensive flow waveforms and then used these waveforms as flow boundary conditions in blood flow simulations for each virtual patient. The total size of the virtual cohort was then $2 \times 82 = 164$ virtual patients with 82 distinct anatomies, each with two systemic flow phenotypes. Since hypertension was present in approximately 50% of the cases in the reference clinical trials (Table 1), this 50/50 distribution in the FD-PASS allowed meaningful subgroup analyses to understand how hypertension affected the post-operative occlusion of aneurysms.

**Virtual cohort size and power calculation**. When designing in-silico trials, the sample size and variability (e.g., in anatomies) of the underlying patient cohort should be such that it reflects real life. The sample size should be large enough to enable to detect relevant effects of the order of magnitude expected in conventional clinical trials. Regarding in-silico patient variability, the source cohort characteristics must be evaluated against the corresponding distributions from real trials. The FD-PASS in-silico trial was a single-arm study. Its primary endpoint was AMVR >35% as a surrogate for complete occlusion of the aneurysm after flow diversion. Based on data available in the literature, we formulated these hypotheses: 85% of patients with wide-necked aneurysms will have complete occlusion in the group treated using the FD; 70% of patients with wide-necked aneurysms will have complete occlusion in the group treated using the conventional techniques (e.g., coiling, surgical clipping, etc.)[36,37]. Considering a type I error of 0.05 and a power of 90%, the number of subjects required to demonstrate a significant difference of 15% between the study group (flow diversion) and the literature-based efficacy of the conventional techniques was 82 patients. Considering a discard rate of 10%, a total of 90 virtual patients should be included.

**Flow-diverter virtual device model**. In the FD-PASS trial, digital models of PEDs, consisting of 48 wires with a 30-micrometre thickness were created and deployed in the parent vessel using a fast virtual stent (FVS) placement method. The FVS method was validated based on in-vitro experiments in[38]. We followed the guidelines presented in[39,40] to select the PED. Hence, the fully-expanded device diameter was equal to that of the proximal parent vessel and the maximal device expansion at the neck was ensured. We focused on the flow diversion efficacy of the PED stent and techniques used to improve the local stent porosity, e.g., the push-pull technique[41], or wall apposition were not considered. A porosity of 70% at the neck is reported to be optimal for aneurysm occlusion[42,43]. This is not achievable in practice due to oversizing, device deformation, manipulation or aneurysm location[40,42]. Multi-device constructs can achieve the favourable porosity, although it increases the risk of ischaemic complications[39,42,44]. A retrospective study of 523 PED treatments proved the aneurysm neck metal coverage necessary for the successful occlusion of aneurysms[17]. However, in this study, no significant association was found between the number of devices per patient and the aneurysm occlusion rate. In another retrospective study of 178 PED treatments, Chalouhi et al.[45] found similar occlusion rates but more complications in aneurysms treated with multiple devices compared to those treated with a single device. Zhang et al.[46] used computational simulations and showed that a porosity of around 70% is sufficient to induce haemodynamic stasis required for sac thrombosis. Accordingly, we remark that, considering haemodynamic performance of FDs, number of PEDs used is of less relevance if an adequate neck metal coverage is achieved. In FD-PASS, all virtual patients were treated with a single device. However, we controlled the aneurysm neck coverage and ensured good stent porosity that was comparable to that in clinical practice. The mean porosity at the neck averaged over the virtual population was $73.2\% \pm 3.8\%$ with minimum and maximum mean porosities of 66.6% and 82.6%, respectively. The virtual PED porosity values were in accordance with those reported in clinical studies (e.g.,[42]) and other computational studies (e.g.,[38]). Since we were only interested in the effect of FDs on the intra-aneurysmal flow, the FD models were clipped to reduce the computational costs. Maintaining portions covering the aneurysm neck, portions of the FDs laying entirely on the vessel wall were removed. The effect of partial stent modelling on intra-aneurysmal haemodynamics was found to be negligible in a previous study[47].

**Virtual patient anatomies**. Retrospective patient-specific imaging data from the @neurIST project (Integrated Biomedical Informatics for the Management of Cerebral Aneurysms;[9]) were used to create the FD-PASS virtual patient cohort. The @neurIST project was a 4-year project initiated in 2006 that comprised 28 public and private institutions from 12 European countries. Within the project, clinical and imaging data were collected from nearly 500 patients with ruptured and unruptured aneurysms. As part of the @neurIST processing toolchain, anatomic surface models were obtained for over 300 aneurysms from 3D rotational angiography images using an automatic segmentation method based on geodesic active regions. Details of the processing toolchain and the verification and validation of the techniques used to generate the @neurIST cohort of virtual anatomies are reported in[48,49]. A population of 301 patient-specific simulation-ready surface models of saccular intracranial aneurysms from the @neurIST project was the reference population in this study. In FD-PASS, aneurysm dimensions were measured on the virtual anatomies obtained from 3D rotational angiography images.

For each virtual anatomy, the inlet branches were truncated at the beginning of the ICA cavernous segment and extruded by an entry length of 5× the inlet diameter to allow for fully developed flow. Outlet branches were automatically clipped 20 mm after their proximal bifurcation. Branches shorter than 20 mm were extruded before truncation. Volumetric meshes were generated using ANSYS ICEM CFD v19.1 (Ansys Inc., Canonsburg, PA, USA). Tetrahedral elements with a maximum edge size of 0.2 mm and five layers of prismatic elements with a maximum edge size of 0.1 mm were used to discretise the core region of the computational domain. Where the PED was present, a maximum edge size of 0.01 mm was set on the wires.

**Virtual physiology flow conditions**. The processed virtual anatomies were then combined with the virtual physiology to generate virtual patients, who were treated with virtual devices and used for blood flow or thrombosis simulations. Unsteady blood flow simulations were solved in ANSYS CFX v19.1 (Ansys Inc., Canonsburg, PA, USA) using a finite volume method. Arterial wall distensibility was not considered (rigid-wall assumption). Time-varying inflow boundary conditions were imposed based on the normotensive and hypertensive waveforms generated for each virtual anatomy (details are provided below). A Poiseuille profile was enforced onto the cross-sectionally averaged flow rate given by the patient-specific inlet conditions. In blood flow simulations with no thrombosis simulation, blood was considered incompressible and Newtonian with a density of $1066 \text{ kg} \cdot \text{m}^{-3}$ and dynamic viscosity of $0.0035 \text{ Pa} \cdot \text{s}$. The cardiac cycle was discretised in time into 200 equal steps. The time-step size was set according to the @neurIST processing toolchain; time-step size independence tests were performed, as described in[49,50]. The validity of the above techniques and assumptions was studied in[50–52] and more overviewed in[53]. Simulations were run for three cardiac cycles and results from the last cycle were used in the analyses, to reduce the effects of initial transients. Mesh independence tests were performed for both blood flow and thrombosis simulations and reported in[27].

For each virtual anatomy in the FD-PASS cohort, we simulated two physiological flow conditions, i.e., a normotensive condition and a hypertensive condition, using models of cerebral autoregulation and by imposing two different inflow boundary waveforms at the ICA inlet boundary. Normotensive ICA flow waveforms were taken as the mean of a virtual population of waveforms generated by a Gaussian process model (GPM) trained on patient-specific phase-contrast magnetic resonance imaging measurements of ICA flow in 17 healthy young adults (age $= 28 \pm 7$ years). Details on this GPM and its validation are reported in[35,54]. To maintain a physiologically realistic flow specific to each virtual patient, Poiseuille's

law was used to scale the GPM-generated waveforms such that the time-averaged wall shear stress was the physiological value of 1.5 Pa at the ICA inlet. While the definition of hypertension has often focused on the elevation of diastolic blood pressure, it has been shown that systolic blood pressure (SPB ≥140 mmHg) is superior when defining hypertension and the associated cardiovascular risks[55]. To generate hypertensive flow waveforms, we used a model of intra-subject flow variability that incorporated a mathematical description of the cerebral autoregulation system. The model took the normotensive flow waveform calculated for each virtual anatomy, estimated its associated normotensive pressure waveform, scaled the pressure waveform to a hypertensive pressure waveform (SPB ≥140 mmHg) and finally calculated the associated hypertensive flow waveform. The mean age of the virtual population used in FD-PASS trial was 54.6 ± 9.62 years. However, the mean age of the population we used for modelling normotensive flow conditions was 28 ± 7. When producing hypertensive flow conditions for each virtual anatomy, the dependency on age was modelled using the methods detailed in[54]. However, the effect of age on flow diversion outcomes has previously been hypothesised to be related to inadequate endothelialisation in elderly individuals[16,17], which was beyond the scope of the FD-PASS trial. Therefore, we argue that age differences between the anatomical and normotensive physiological populations had a minimal effect on the FD-PASS findings.

**Virtual thrombus formation**. A thrombus formation model was applied in a subset of virtual patients to better understand the possible causes of device failure. This model was used to distinguish between two types of thrombi that can form inside the aneurysm post-treatment: white thrombi, which are rich in fibrin and platelets, and red thrombi. Red thrombi are characterised by (i) fewer enmeshed platelets, (ii) not facilitating the formation of a neointimal layer, (iii) being prone to continuous fibrinolysis and renewal and (iv) inducing autolytic activities in the wall resulting in the weakening of the wall and ultimately rupture. Non-organised red thrombi after flow diversion have been suggested as a potential predictor for post-treatment rupture[56]. We previously developed and validated a coupled flow and thrombosis model[27] capable of predicting the type of thrombus and the resulting stability of the clot that forms after flow diversion. The bulk initiation and propagation mechanisms for thrombosis were assumed to be due to flow stasis only, and initiation due to vessel wall injury or high platelet shear was not modelled. After placing a FD, thrombosis was assumed to initiate in regions with a time-averaged shear rate (SR) below $SR_t = 25s^{-1}$[57] and residence time (RT) greater than $RT_t = 5s$[58]. The model included five biochemical species: prothrombin, thrombin, anti-thrombin, fibrinogen, fibrin, and three categories of platelets (resting platelets, activated platelets and fibrin bound aggregated platelets). Four biochemically coupled events were modelled to simulate the formation of a clot consisting of a fibrin mesh and aggregated platelets. (i) Thrombin generation occurred by the conversion of prothrombin to thrombin on the surface of resting and activated platelets, where the reaction kinetics were faster for activated platelets. Thrombin inhibition by its primary plasma inhibitor (anti-thrombin) in the absence of heparin catalysis was also considered. (ii) Fibrin generation occurred in the presence of thrombin, which converted fibrinogen to fibrin monomers. We did not consider further polymerisation of the fibrin monomers. (iii) Platelet activation occurred when resting platelets became activated by exposure to thrombin or other activated platelets. The latter mechanism was a surrogate for activation by agonists released from other activated platelets[59]. (iv) Platelet aggregation in the presence of fibrin occurred when platelets attached to the fibrin network aggregated to form bound platelets. Bound platelets were assumed to activate prothrombin and other resting platelets. The governing equations for the coupled system of blood flow and post-flow diversion thrombosis were reported in[27], in which the model was validated with in-vitro experiments. The momentum and transport equations for biochemical species were solved in ANSYS CFX v19.1 (Ansys Inc., Canonsburg, PA, USA) using a finite volume method. Simulations of the coupled flow and thrombosis were run for 400 s of simulation time using CFX's adaptive time-stepping with the minimum, maximum, and initial time-steps of 0.0001, 0.05 and 0.01 s. Solutions at each time step converged when the maximum residual of the computational domain was less than $5 \times 10^{-4}$.

To quantitatively measure the stability of the intra-aneurysmal thrombus, we defined the FiPi as the relative difference of the platelet concentration between a closed (i.e., a chamber with no inlet/outlet) system and an open system:

$$\mathcal{P} = \frac{C_{bp}^{open} - C_{bp}^{closed}}{C_{bp}^{closed}} = \frac{C_{bp}^{open}}{C_{rp,0} + C_{ap,0}} - 1. \qquad (1)$$

In Eq. (1), $C_{rp,0}$ and $C_{ap,0}$ are the initial concentrations of resting and activated platelets in clot-free blood, respectively. FiPi quantified the effect of blood flow on the transport of platelets to and from the site of clot formation and on the final platelet content of the formed clot.

**Definition of the primary endpoint**. Clinical trials evaluating devices for the treatment of aneurysm typically use endpoints such as neurological morbidity and mortality, intracranial haemorrhage and ischaemic stroke as measures of safety; a typical efficacy endpoints is the angiographic occlusion rate measured using the Raymond-Roy 5-point occlusion scale at 180 days and 1 year. Since a major

purpose of in-silico trials is to help improve the design of an investigational device before it goes to clinical safety trials in humans, the FD-PASS focused solely on device efficacy as the primary endpoint.

While angiographic occlusion immediately post-intervention can be simulated in the computational model, the simulation of long-term post-operative outcomes (e.g., at 1 year) is not feasible due to the limited understanding of the long-term physiological response. Several groups have performed computational fluid dynamics studies of cerebral aneurysms[10,60,61] and suggested the use post-flow diversion aneurysm mean velocity as a proxy haemodynamic metric for predicting aneurysm occlusion. In[10], it was demonstrated in 12 patients who had a reduction of over 35% in the aneurysm mean velocity resulted in aneurysm occlusion with 99% sensitivity. Therefore, we focused on one primary endpoint, the post-treatment AMVR, as a surrogate for angiographic occlusion and set an AMVR >35% as the data-independent criterion for flow diversion success.

**Reference human clinical trials**. The PUFS[29–31] was a prospective multi-centre, single-arm, trial conducted between 2008 and 2014. Patients were included if they had a large/giant ICA aneurysm (petrous through superior hypophyseal segments) larger than 10 mm in diameter and with a neck larger than 4 mm. A total of 108 patients with 108 aneurysms were enrolled and treated with PED in ten centres. The efficacy endpoint of the study was complete angiographic occlusion of the aneurysm without major (>50%) stenosis of the parent artery after 180 days and 1, 3, and 5 years. Occlusion rates were approximately 75–95% at 6 months to 5 years.

The ASPIRe[7] was a prospective single-arm, multi-centre observational study. It evaluated all patients who consented and were treated with PED over 3 years in 28 centres. A total of 191 patients with 207 aneurysms were enrolled. The efficacy endpoint of the study was complete angiographic occlusion of the aneurysm at the final follow-up. Occlusion rates were approximately 75% with a median follow-up time of 7.8 months.

The PREMIER[8] was a prospective single-arm, multi-centre trial conducted between 2014 and 2016. Patients were included if they had a small-/medium-sized ICA or vertebral artery aneurysm measuring ≤12 mm in diameter, with neck ≥4 mm or a dome to neck ratio ≤1.5. A total of 141 patients with 141 aneurysms were enrolled and treated with PED in 23 centres. The efficacy endpoint of the study was complete angiographic occlusion of the aneurysm without major (>50%) stenosis of the parent artery after 1 year. Occlusion rates were approximately 77% at 1 year.

**Statistical analysis**. Continuous variables are presented as the number of observations and the mean and standard deviation or the median, minimum and maximum values. We used an unpaired Student's $t$-test to compare continuous variables. Categorical variables are presented using frequency distributions and cross-tabulations. We used contingency table analyses ($\chi^2$ tests) to compare categorical variables. To evaluate the factors associated with the risk of incomplete occlusion (i.e., FD device failure), RRs and CIs were calculated using standard statistical methods[62].

**Reporting summary**. Further information on research design is available in the Nature Research Reporting Summary linked to this article.

## Data availability

The imaging data that support the findings of this study are available from the Centre for Computational Imaging & Simulation Technologies in Biomedicine (www.cistib.org) at the University of Leeds as coordinators of the @neurIST Consortium (www.aneurist.org). Restrictions apply to the availability of these data, which were used under a Consortium Agreement for the current study, and so are not publicly available. Bona fide researchers can request access to the anatomical models and computational models via our collaborative cloud-based research framework MULTI-X (www.multi-x.org). Researchers can register and request access to research data and models developed by our and other groups. Simulations pipelines can be launched on virtual machines using cloud-based computing. Contact A. F. F. (a.frangi@leeds.ac.uk) for collaboration requests, which will be answered within a week. Costs may be involved for cloud storage and execution. Source Data are provided with this paper.

## Code availability

All computational models used in this study have been described in previously published works. The digital device implantation model is described in[38]. A standard Navier–Stokes equation solver in ANSYS CFX v19.1 (Ansys Inc., Canonsburg, PA, USA) was used to solve the flow problems. The clotting model is described in[27] and the same model parameters were used in this study. The flow model uses standard CFX solvers. Reference implementations of the device implantation model and clotting model are available from the authors upon a reasonable collaboration request. Contact A. F. F. (a.frangi@leeds.ac.uk) for requests, which will be answered within a week.

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

## Acknowledgements

A.F.F. acknowledges support from the Royal Academy of Engineering under the RAEng Chair in Emerging Technologies (CiET1919/19) scheme. A.F.F. acknowledges seminal funding from the European Commission to @neurIST 'Integrated Biomedical Informatics for the Management of Cerebral Aneurysms' (FP6-2004-IST-4-027703) and the @neurIST Consortium. A.S.F., T.L. and A.F.F. were funded by the European Commission via InSilc 'In-silico trials for drug-eluting BVS design, development and evaluation' (H2020-SC1-2017-CNECT-2-777119). The Engineering and Physical Sciences Research Council (EPSRC) Centre of Doctoral Training in Fluid Dynamics (EP/L01615X/1) supported M.M. and J.A. The information contained herein reflects only the authors' view, and none of the funders are responsible for any use that may be made of it. We acknowledge support from ANSYS through an Academic Partnership agreement (#1122349).

## Author contributions

A.F.F. established the study concept, and contributed to virtual stenting and image processing methods and access to retrospective data sets. A.S.F. implemented the methods. A.S.F., T.L. and A.F.F. designed the experiments and together with J.V.B. interpreted the results. K.C.B.R. confirmed the virtual trial design and comparative statistical methods and analyses. A.S.F., M.M. and J.A. performed the experiments. A.S.F. and T.L. performed the analyses and with A.F.F. wrote the manuscript. All authors provided critical feedback on the manuscript.

## Competing interests

J.V.B. is a consultant for MicroVention Inc. and Oxford Endovascular Ltd, with an equity interest in the latter.
