## [Peer Review File · Nature Communications]

Reviewers' Comments:

Reviewer #1:

The authors report a computational pipeline designed to evaluate the efficacy of flow diverter stents used to treat cerebral aneurysms. In a virtual environment, diverters are inserted into a diseased artery so that the metal wire of the diverter covers the neck of the aneurysm. Thereafter, the increased resistance associated with the diverter reduces blood flow to the aneurysm sac where clots are formed in the stagnating blood. The goal of this treatment is to detach the aneurysm from the circulation and thus mitigate the risk of rupture. To describe the relevant processes, the authors employed the following validated models:

1. 3D triangulated surface mesh capturing a patient-specific aneurysm and the surrounding vasculature.
2. Simplex deformable stent model utilising a dynamic system approach to capture deployment dynamics.
3. Computational fluid dynamics model of blood flow based on the incompressible Navier-Stokes equations describing Newtonian fluids. This model includes optional porous regions to handle fluid flow inside blood clots.
4. Thrombus formation model based on advection-diffusion-reaction equations describing the evolution of the most important chemical species.

The authors consider 82 aneurysm geometries and 2 flow conditions (normotensive and hypertensive) so that a cohort of 164 virtual patients is investigated. Firstly, they aim to validate their model by comparing the computed efficacy with existing results from clinical trials. Secondly, they use simulations to reflect on hypotheses regarding the failure of flow diverters (i) in patients with a branch artery originating from the aneurysm; (ii) in patients with large/giant/high aspect ratio aneurysms; (iii) in hypertensive patients. Furthermore, they explored scenarios when failure can lead to haemorrhagic or ischaemic stroke.

The analyses are carried out carefully and the corresponding results are presented logically. The topic of the paper is relevant for the medical and physiological modelling community and could be interesting for the readers of *Nature Communications* with generic background. Nevertheless, some major modifications, several minor amendments, and in general a thorough proofreading are recommended before publication.

Major issues

1; I found the endpoint comparison between the current in silico trials (FD-PASS) and the clinical trials (PUFS, ASPIRe, PREMIER) presented in Table 2 rather weak. The authors use the aneurysm spatiotemporal mean velocity reduction (AMVR) as a surrogate for complete occlusion. This single-variable surrogate suggested by Ouaed *et al.* based on 12 CFD simulations of patient-specific aneurysms has a sensitivity of about 99% and a specificity of about 67%. Considering the virtual cohort of the present study, 41.5% of the patients had a side branch originating from the sac which is known to decrease occlusion chance. A more careful evaluation of the endpoint estimation is essential. Uncertainty quantification and testing of alternative methods based on multiple haemodynamic and/or geometric variables is strongly encouraged (see for example Paliwal et al.)

2; The properties of the virtual population used in this study appear to be representative of the cohorts of clinical trials, except the number of PEDs used. The authors attempt to justify their choice based on Maragkos *et al.* but the results of this single clinical study might seem contradictory regarding the currently employed endpoint surrogate: applying multiple PEDs increases the hydrodynamic resistance of the neck and should therefore increase AMVR which

would impact the outcome of the present *in silico* clinical trial. Further justification/explanation is required here depending on the resolution of the first major issue.

3; Whereas I found every information in the manuscript and the cited papers required to understand this study, the readers could benefit greatly from a figure summarising the key steps and models involved in the proposed *in silico* clinical trial. Pictograms and small images visualising the most important elements of the trial would be very helpful. If the authors could extend the supplementary material with concise descriptions of the models (geometry, stent, blood flow, thrombus formation) including governing equations and boundary conditions where appropriate, that would be welcome. Thereafter, I would recommend a shorter description of these models in the method section of the main text.

Minor issues

- The abstract claims that 82 virtual patients were analysed which is inconsistent with 164 patients in the main text.
- “These findings demonstrate for the first time that in-silico trials of medical devices can (i) replicate findings of conventional clinical trials and (ii) incorporate virtual experiments that are impossible in conventional trials.” This claim in the abstract, and later in the introduction is an unnecessary and misleading exaggeration. *In silico* trials have been reported before (for example, Carlier et al.) Please detail the true significance of this study for the medical community instead.
- The referencing style does not follow any of the well-established standards.
- The first 5 sentences of the Introduction are too long, and hard to penetrate through, please rephrase and restructure.
- “Pipeline Embolisation Device (PED) the Surpass are currently FDA approved”, awkwardly phrased and the verb should be singular anyway.
- “no other FD have been”  no other FDs have been OR no other FD has been
- “FDs are the primary used to treat uncoilable”  FDs are primarily used to...
- “represent a small fraction all intracranial aneurysms”  represent a small fraction of intracranial aneurysms
- It is recommended to save space in the main text by avoiding double-presenting information already available in tables. For instance, detailing population statistics on page 4 could be simply replaced with something like “Population statistics are summarised in Table 1. Properties of the virtual cohort are statistically representative of patient groups recruited in clinical trials”. The same concept should be applied later in the text.
- “simulation results clearly show that a side branch is likely to cause higher residual flow”  is it clearly shown or is it likely? Consider “simulation results indicate that a side branch is likely to cause ...”
- I could not find AR (aspect ratio) introduced in the text as an abbreviation.
- STAV is almost never used without AMVR and it appears to be highly correlated with MTAV. This should be mentioned in the text. Thereafter, it seems sufficient to present STAV and MTAV statistics solely in the supplementary materials. For these reasons, I would suggest dropping both STAV and MTAV from the main text and keeping only AMVR. The manuscript is quite long and already a bit hard to read here and there

because of these abbreviations. At the end of the day, only AMVR is used for endpoint prediction (which should be re-evaluated).

- The resolution of Figure 1 is poor, please improve it and ensure that the red boxes on the LHS are rotated by 180° so that the text alignment is consistent.
- “we observed the rapid formation of a non-organised red thrombus (FiPi < 0.15) aneurysm bleb”  awkward sentence, please rephrase.
- Replace “yr” with year everywhere.
- Figure 2. “Time values are shown in units of simulation time” what does this mean and why is it not possible to present results based on wall time?

Reviewer #2:

Remarks to the Author:

In this paper, the authors presented the results of a flow diverter performance assessment in-silico trial using CFD and compared the results to 3 clinical trials with the Pipeline Embolization Device. They concluded that the in-silico trials replicated the findings of the clinical studies and in addition allowed broader investigation of factors that will be hard to study in a conventional trial (eg. the effect of blood pressure on occlusion, the effect of morphology on post-treatment hemorrhage, and the effect of blood pressure on post-treatment ischemic events). Moreover, such exploratory analyses can also explain findings rather than just making the associations. For example, the modeling suggests the formation of unstable red thrombus and in-flow jet into the aneurysms with complex morphology is the reason behind the increase in rupture risk. Although the results are replicable of other studies, I wonder if the authors can provide more granular data and comment on its predictive ability in the @neuroIST population. For example, did the aneurysms that were predicted to occlude by CFD actually occluded? How about the ones that were predicted to have a thromboembolic event or rupture after treatment? What were their clinical outcomes. Secondly, not all results are congruent between in-silico trials and clinical observations. For example, the Insilco trials suggest that giant and small aneurysms have no difference in occlusion rates, but clinical data suggest that giant aneurysms may have a higher and faster occlusion rates. The main weakness of such methods is that it cannot account for factors rather than hemodynamics that may affect occlusion outcomes although no doubt that it represents a novel and powerful way to understand the hemodynamic mechanisms of flow diversion.

Point-by-point Response to Reviewers

Reviewer #1

The authors report a computational pipeline designed to evaluate the efficacy of flow diverter stents used to treat cerebral aneurysms. In a virtual environment, diverters are inserted into a diseased artery so that the metal wire of the diverter covers the neck of the aneurysm. Thereafter, the increased resistance associated with the diverter reduces blood flow to the aneurysm sac where clots are formed in the stagnating blood. The goal of this treatment is to detach the aneurysm from the circulation and thus mitigate the risk of rupture. To describe the relevant processes, the authors employed the following validated models:

- 1. 3D triangulated surface mesh capturing a patient-specific aneurysm and the surrounding vasculature.*
- 2. Simplex deformable stent model utilising a dynamic system approach to capture deployment dynamics.*
- 3. Computational fluid dynamics model of blood flow based on the incompressible Navier-Stokes equations describing Newtonian fluids. This model includes optional porous regions to handle fluid flow inside blood clots.*
- 4. Thrombus formation model based on advection-diffusion-reaction equations describing the evolution of the most important chemical species.*

The authors consider 82 aneurysm geometries and 2 flow conditions (normotensive and hypertensive) so that a cohort of 164 virtual patients is investigated.

Firstly, they aim to validate their model by comparing the computed efficacy with existing results from clinical trials. Secondly, they use simulations to reflect on hypotheses regarding the failure of flow diverters (i) in patients with a branch artery originating from the aneurysm; (ii) in patients with large/giant/high aspect ratio aneurysms; (iii) in hypertensive patients. Furthermore, they explored scenarios when failure can lead to haemorrhagic or ischaemic stroke.

The analyses are carried out carefully and the corresponding results are presented logically. The topic of the paper is relevant for the medical and physiological modelling community and could be interesting for the readers of Nature Communications with generic background. Nevertheless, some major modifications, several minor amendments, and in general a thorough proofreading are recommended before publication.

Major issues

R1.1) I found the endpoint comparison between the current in silico trials (FD-PASS) and the clinical trials (PUFS, ASPIRe, PREMIER) presented in Table 2 rather weak. The authors use the aneurysm spatiotemporal mean velocity reduction (AMVR) as a surrogate for complete occlusion. This single-variable surrogate suggested by Ouared et al. based on 12 CFD simulations of patient-specific aneurysms has a sensitivity of about 99% and a specificity of about 67%. Considering the virtual cohort of the present study, 41.5% of the patients had a side branch originating from the sac which is known to decrease occlusion chance. A more careful evaluation of the endpoint estimation is essential. Uncertainty quantification and testing of alternative methods based on multiple haemodynamic and/or geometric variables is strongly encouraged (see for example Paliwal et al.)

FD-PASS uses a virtual population but is not a *virtual twin trial* to *validate* an in-silico trial framework for evaluating flow-diverters or to *confirm* findings of a clinical trial in a head-to-head manner (see Figure below for reference). Instead, in FD-PASS is a *virtual chimera trial*, where we aimed to demonstrate that results from clinical trials could be *in silico replicated* on an equivalent virtual population but fully independent to that of the corresponding clinical trial. For *replication*, independence of the cohorts is stronger. Regarding the endpoints, we used a *technical endpoint* (i.e., angiographic occlusion represented by aneurysm mean velocity reduction (AMVR) > 35%) as a surrogate for a clinical endpoint (complete aneurysm occlusion). We did not aim to devise a pre-operative predictor of flow-diversion success; and, therefore, training a predictive model on real cases with known outcomes and evaluating model sensitivity/uncertainty was out of the scope of this study. In FD-PASS, we used technical endpoints that were obtained from completely *independent* populations and demonstrated that we can replicate findings from clinical trials based on these technical endpoints. Cebral et al. (2019) simulated haemodynamics in 36 aneurysms treated with flow diverters, albeit with intra-saccular devices, and showed that AMVR > 20% and Inflow reduction (INFR) > 30% were accurate predictors of successful occlusion. Applying these criteria to our cohort resulted in occlusion rates of 78% and 83% which agreed with occlusion rates reported by the clinical trials we attempted to replicate.

Virtual Populations, Twins & Chimeras

Figure 1: Illustration of Virtual Populations of Twins and Chimeras. NB: Example given for another device but used here only to illustrate the meaning of the terms twin and chimera.

Discussion, end of 5th the paragraph: FD-PASS estimation of occlusion rate was performed based on one primary endpoint, the post-treatment AMVR > 35% as a surrogate for angiographic occlusion, which was obtained by computational simulation of an independent cohort of $N = 12$ flow diverted aneurysms (Ouaed et al. 2016). While we emphasise that FD-PASS did not aim to devise or validate a predictive surrogate endpoint, we explored what happens to the occlusion rates when we set a different surrogate endpoint. We examined AMVR > 20% and post-treatment inflow reduction (INFR) at the aneurysm neck $\geq 30\%$, which were presented as accurate simulation-based predictors of flow diversion success in a cohort of $N = 36$ patients treated with intrasaccular flow diverters (Cebral et al., 2019).

Occlusion rates of 78% and 83% were achieved when we used AMVR > 20% and INFR > 30% as primary endpoints, respectively. This demonstrated that the estimated occlusion rates were not sensitive to the choice of surrogate endpoint.

R1.2) The properties of the virtual population used in this study appear to be representative of the cohorts of clinical trials, except the number of PEDs used. The authors attempt to justify their choice based on Maragkos et al. but the results of this single clinical study might seem contradictory regarding the currently employed endpoint surrogate: applying multiple PEDs increases the hydrodynamic resistance of the neck and should therefore increase AMVR which would impact the outcome of the present in silico clinical trial. Further justification/explanation is required here depending on the resolution of the first major issue.

A porosity of 70% at the neck is reported to be optimal for aneurysm occlusion (Jou et al. 2016; Ravindran et al. 2020). However, this is not usually achievable due to oversizing, device deformation, manipulation, or aneurysm location (Jou et al. 2016; Shapiro et al. 2014). Use of multiple devices was suggested to alleviate this issue and provide sufficient metal coverage and, thus, flow diversion. In our study, we already controlled the final metal coverage of the neck to be around 70% (i.e., the target coverage when single/multiple devices are used in the clinical practice). Thus, although we agree with the reviewer that use of multiple PEDs will increase the haemodynamic resistance of the neck and will affect AMVR, we argue that the haemodynamic behaviour of flow diverters in clinical trials we compared against could be replicated using a single device as long as the final metal coverage meets the criterion clinically considered to be adequate for aneurysm occlusion (65%-75%).

In fact, while in the PUFs trial most patients had more than one FD implanted, both ASPIRe and PREMIER trials only implanted two more FDs in less than 20% of patients. Yet the 6–12-month angiographic occlusion rates were similar across all three studies, 73–77%. This provides evidence that if the neck porosity requirement of 75% is met, implanting additional devices does not contribute positively to the occlusion probability.

Methods, FD virtual device model: [...]. A porosity of 70% at the neck is reported to be optimal for aneurysm occlusion (Jou et al. 2016; Ravindran et al. 2020). This is not achievable in practice due to oversizing, device deformation, manipulation, or aneurysm location (Jou et al. 2016; Shapiro et al. 2014). Multi-device constructs can achieve the favourable porosity, although it increases the risk of ischaemic complications (FDA 2011; Jou et al. 2016; Shapiro et al. 2012). A retrospective study of 523 PED treatments proved the aneurysm neck metal coverage necessary for the successful occlusion of aneurysms (Maragkos et al. 2019). However, in this study, no significant association was found between the number of devices per patient and the aneurysm occlusion rate. In another retrospective study of 178 PED treatments, Chalouhi et al. (2014) found similar occlusion rates but more complications in aneurysms treated with multiple devices compared to those treated with a single device. Zhang et al. (2019) used computational simulations and showed that a porosity of around 70% is sufficient to induce haemodynamic stasis required for sac thrombosis. Accordingly, we remark that, considering haemodynamic performance of FDs, number of PEDs used is of less relevance if an adequate neck metal coverage is achieved. In FD-PASS, all virtual patients were treated with a single device. However, we controlled the aneurysm neck coverage and ensured good stent porosity that was comparable to that in clinical practice. The mean porosity at the neck averaged over the virtual population was $73.2\% \pm 3.8\%$ with minimum and maximum mean porosities of 66.6% and 82.6%, respectively. The virtual PED porosity values were in accordance with those reported in clinical studies (e.g., Jou et al. 2016) and other computational studies (e.g., Larrabide et al. 2012). [...].

3) Whereas I found every information in the manuscript and the cited papers required to understand this study, the readers could benefit greatly from a figure summarising the key steps and models involved in the proposed in silico clinical trial. Pictograms and small images

visualising the most important elements of the trial would be very helpful. If the authors could extend the supplementary material with concise descriptions of the models (geometry, stent, blood flow, thrombus formation) including governing equations and boundary conditions where appropriate, that would be welcome. Thereafter, I would recommend a shorter description of these models in the method section of the main text.

We have added a figure to concisely explain the key steps of the virtual in-silico trial. As for the details of the computational models, this paper focuses on the in-silico trial and claims no new contributions to the computational models used. As the models have been previously detailed in other publications, we believe a reference to each model is sufficient rather than reproducing previously published materials.

The abstract claims that 82 virtual patients were analysed which is inconsistent with 164 patients in the main text.

Abstract: Replace in sentence in question with “164 virtual patients with 82 distinct anatomies”.

“These findings demonstrate for the first time that in-silico trials of medical devices can (i) replicate findings of conventional clinical trials and (ii) incorporate virtual experiments that are impossible in conventional trials.” This claim in the abstract, and later in the introduction is an unnecessary and misleading exaggeration. In silico trials have been reported before (for example, Carlier et al.) Please detail the true significance of this study for the medical community instead.

We appreciate there have been in-silico trials performed before ours, although we are not aware of previously published in-silico trials for endovascular devices. Nevertheless, we have made the claims more specific to the in-silico trial at hand. Abstract: “These findings demonstrate for the first time that in-silico trials of endovascular medical devices can (i) replicate findings of conventional clinical trials and (ii) incorporate virtual experiments and sub-group analyses that are difficult or impossible in conventional trials and discover new insights on treatment failure, e.g., in the presence of side-branches or hypertension.”

The referencing style does not follow any of the well-established standards.

The bibliography now follows the Nature Communications style.

The first 5 sentences of the Introduction are too long, and hard to penetrate through, please rephrase and restructure.

We rewrote the part in question for clarity. First paragraph of Introduction: “Recent developments in patient-specific computational simulations have enabled in-silico trials to predict the safety and efficacy of novel drugs, medical devices, or other treatments as part of the research and development (R&D) [1,2]. Some of the benefits of in-silico trials include: i) Enabling more evidence to be obtained before bench or animal studies are started; ii) Extending the trial cohort to rare, extreme or difficult-to-recruit patient phenotypes; iii) Directly comparing two alternative treatments in the same virtual population (resulting in a reduction in the variance in the observed effect); iv) Evaluating devices under practically challenging physiological conditions that could represent extreme but plausible applications (off-label use); v) Reducing the number of animals and humans required in trials, and the refinement of long-term studies to minimise suffering [3]. Drug and device regulatory authorities, such as the FDA, are working with the biomedical modelling and simulation community to specify the

requirements for introducing evidence obtained from in-silico trials into the regulatory process or informing the design of conventional trials [4].”

“Pipeline Embolisation Device (PED) the Surpass are currently FDA approved”, awkwardly phrased and the verb should be singular anyway.

Third paragraph of Introduction: “Although multiple types of FDs are available in Europe, only the Pipeline Embolization Device (Medtronic) and the Surpass Streamline Flow Diverter (Stryker) are currently FDA approved in the United States [32].”

“no other FD have been”  no other FDs have been OR no other FD has been

Third paragraph of Introduction: “...no other FDs have been shown to outperform the PED...”.

“FDs are the primary used to treat uncoilable”  FDs are primarily used to...

Fourth paragraph of Introduction: “FDs are primary used to treat uncoilable...”.

“represent a small fraction all intracranial aneurysms”  represent a small fraction of intracranial aneurysms

Fourth paragraph of Introduction: “...large/giant aneurysms represent a small fraction of all intracranial aneurysms...”.

It is recommended to save space in the main text by avoiding double-presenting information already available in tables. For instance, detailing population statistics on page 4 could be simply replaced with something like “Population statistics are summarised in Table 1. Properties of the virtual cohort are statistically representative of patient groups recruited in clinical trials”. The same concept should be applied later in the text.

Modified to reduce repetition: “Population statistics and measurements of the aneurysms are provided in Table 1 along a comparison of the in-silico cohort vs. other trial cohorts from the literature.”

“simulation results clearly show that a side branch is likely to cause higher residual flow”  is it clearly shown or is it likely? Consider “simulation results indicate that a side branch is likely to cause ...”

Fourth paragraph of Results: “The virtual flow simulation indicates that a side branch is likely to cause higher residual flow within the aneurysm sac.”

I could not find AR (aspect ratio) introduced in the text as an abbreviation.

Third paragraph of Results: “...aspect ratio (AR)...”

STAV is almost never used without AMVR and it appears to be highly correlated with MTAV. This should be mentioned in the text. Thereafter, it seems sufficient to present STAV and MTAV statistics solely in the supplementary materials. For these reasons, I would suggest dropping both STAV and MTAV from the main text and keeping only AMVR. The manuscript is quite long and already a bit hard to read here and there because of these abbreviations. At the end of the day, only AMVR is used for endpoint prediction (which should be re-evaluated).

In our opinion, STAV is naturally used when defining AMVR so should remain in the main body of the text. Removing MTAV alone would not in our view improve the readability of the manuscript.

The resolution of Figure 1 is poor, please improve it and ensure that the red boxes on the LHS are rotated by 180° so that the text alignment is consistent.

Figure modified for better resolution and matching vertical text alignment.

“we observed the rapid formation of a non-organised red thrombus (FiPi < 0.15) aneurysm bleb”  awkward sentence, please rephrase.

Last paragraph of Results: *“In the more complex case 2, we observed the rapid formation of a non-organised red thrombus (FiPi < 0.15) inside the aneurysm bleb. A persistent flow jet was present in the sac during the entire time course of the clot formation.”*

Replace “yr” with year everywhere.

Changed where appropriate.

Figure 2. “Time values are shown in units of simulation time” what does this mean and why is it not possible to present results based on wall time?

It is not clear what the reviewer means here by “wall time”. The clot formation model in Sarrami-Foroushani *et al.* 2019 is based on biophysically accurate equations and has been previously validated against in-vitro experiments, but the time scales of the reactions in the model are not the same as in real patients. Therefore, the model does not answer questions such as “how long will it take for this aneurysm to occlude” but does provide predictions about the final steady-state configuration of the clot. We included the caveat to make it clear the aneurysms would not occlude in 30 s in practice.

Reviewer #2

In this paper, the authors presented the results of a flow diverter performance assessment in-silico trial using CFD and compared the results to 3 clinical trials with the Pipeline Embolization Device. They concluded that the in-silico trials replicated the findings of the clinical studies and in addition allowed broader investigation of factors that will be hard to study in a conventional trial (eg. the effect of blood pressure on occlusion, the effect of morphology on post-treatment hemorrhage, and the effect of blood pressure on post-treatment ischemic events). Moreover, such exploratory analyses can also explain findings rather than just making the associations. For example, the modeling suggests the formation of unstable red thrombus and in-flow jet into the aneurysms with complex morphology is the reason behind the increase in rupture risk.

Although the results are replicable of other studies, I wonder if the authors can provide more granular data and comment on its predictive ability in the @neurIST population. For example, did the aneurysms that were predicted to occlude by CFD actually occluded? How about the ones that were predicted to have a thromboembolic event or rupture after treatment? What were their clinical outcomes.

The aneurysm cases from @neurIST that were used to build virtual patient models were not treated with flow diverters within that study and were instead treated with coiling. Consequently, we do not have a ground truth outcome nor follow-up imaging for these cases. The patient-specific anatomies

were sufficient for the purpose of FD-PASS, which was to replicate the findings of previous clinical trials, rather than performing a digital twin trial to perform direct validation of treatment outcome predictions.

Secondly, not all results are congruent between in-silico trials and clinical observations. For example, the Insilco trials suggest that giant and small aneurysms have no difference in occlusion rates, but clinical data suggest that giant aneurysms may have a higher and faster occlusion rates.

We point out that, as seen in Tables 4 and 6, the effect of aneurysm size on occlusion rate and risk ratio is statistically significant, but only in the hypertensive sub-group. In fact, this is an example where sub-group analysis sheds additional insights and can only be performed in an *in-silico* trial as controlling for hypertension in a clinical trial is unfeasible.

The main weakness of such methods is that it cannot account for factors rather than hemodynamics that may affect occlusion outcomes although no doubt that it represents a novel and powerful way to understand the hemodynamic mechanisms of flow diversion.

This paper is a counter example of this statement as we account for the biochemistry of the clotting process, i.e. we go beyond haemodynamics. We use a previously validated thrombus formation model that was used in an exploratory analysis to predict the extent of the final stable clot in four virtual cases. There are of course many factors of human biology and physiology that a computer simulation can never hope to capture, but mathematical models do not have to be complete to be useful. We argue a conventional clinical trial has also many limitations and constraints both in its design and implementation (e.g. randomisation to ensure exchangeability, double-blind design, cost, risks on patients, recruitment duration, etc) yet has been used with benefits. FD-PASS demonstrates that, in the context of flow diversion, modelling flow alone already provides useful information to understand where and how clinical trials could be improved to reach better and more robust conclusions, especially in high-risk sub-groups, such as hypertensive patients.

Yours respectfully,

April 19th 2021,

Prof. Alejandro F. Frangi
Corresponding author
University of Leeds & KU Leuven

Reviewers' Comments:

Reviewer #1:

Remarks to the Author:

The authors resolved the raised issues and answered all of my questions. I believe that once the following minor amendments are met, and potentially after another thorough proofreading, the manuscript should be ready for publication in Nature Communications:

-Figure references in the text have not been updated after Fig. 1 was added, see, for example, references in page 5 to Fig. 1 which clearly correspond to Fig. 2, etc.

-page 12, end of the third paragraph, should be "were not sensitive to the choice of surrogate endpoint."

Reviewer #2:

Remarks to the Author:

The authors have addressed the concerns of both reviewers to satisfaction and I recommend acceptance at this point

Point-by-point Response to Reviewers

Reviewer #1

The authors resolved the raised issues and answered all of my questions. I believe that once the following minor amendments are met, and potentially after another thorough proofreading, the manuscript should be ready for publication in Nature Communications:

-Figure references in the text have not been updated after Fig. 1 was added, see, for example, references in page 5 to Fig. 1 which clearly correspond to Fig. 2, etc.

The figure and table references have been corrected.

-page 12, end of the third paragraph, should be "were not sensitive to the choice of surrogate endpoint."

This sentence has been corrected.

Yours respectfully,

May 17th 2021,

Prof. Alejandro F. Frangi
Corresponding author
University of Leeds & KU Leuven